# Polysaccharide Derived from *Nelumbo nucifera* Lotus Plumule Shows Potential Prebiotic Activity and Ameliorates Insulin Resistance in HepG2 Cells

**DOI:** 10.3390/polym13111780

**Published:** 2021-05-28

**Authors:** Bao Le, Pham-Thi-Ngoc Anh, Seung-Hwan Yang

**Affiliations:** 1Faculty of Pharmacy, Ton Duc Thang University, Ho Chi Minh City 700000, Vietnam; lebao@tdtu.edu.vn; 2Department of Biotechnology, Chonnam National University, Yeosu 59626, Korea; 178700@jnu.ac.kr

**Keywords:** HepG2, diabetes, IRS1/PI3K/Akt, *Nelumbo nucifera*, polysaccharide

## Abstract

Polysaccharides are key bioactive compounds in lotus plumule tea, but their anti-diabetes activities remain unclear. The purpose of this study was to investigate the prebiotic activities of a novel polysaccharide fraction from the *Nelumbo nucifera* lotus plumule, and to examine its regulation of glucose metabolism in insulin-resistant HepG2 cells. The *N. nucifera* polysaccharide (NNP) was purified after discoloration, hot water extraction, ethanol precipitation, and DEAE-cellulose chromatography to obtain purified polysaccharide fractions (NNP-2). Fourier transform infrared spectroscopy was used to analyze the main structural characteristics and functional group of NNP-2. Physicochemical characterization indicated that NNP-2 had a molecular weight of 110.47 kDa and consisted of xylose, glucose, fructose, galactose, and fucose in a molar ratio of 33.4:25.7:22.0:10.5:8.1. The prebiotic activity of NNP-2 was demonstrated in vitro using *Lactobacillus* and *Bifidobacterium*. Furthermore, NNP-2 showed bioactivity against α-glucosidase (IC_50_ = 97.32 µg/mL). High glucose-induced insulin-resistant HepG2 cells were used to study the effect of NNP-2 on glucose consumption, and the molecular mechanism of the insulin transduction pathway was studied using RT-qPCR. NNP-2 could improve insulin resistance by modulating the IRS1/PI3K/Akt pathway in insulin-resistant HepG2 cells. Our data demonstrated that the *Nelumbo nucifera* polysaccharides are potential sources for nutraceuticals, and we propose functional food developments from the bioactive polysaccharides of *N. nucifera* for the management of diabetes.

## 1. Introduction

Diabetes is a chronic metabolic disorder caused by various pathological mechanisms such as insulin resistance or insufficient insulin secretion, resulting in increased concentrations of glucose in the blood [1]. The development of insulin resistance is also associated with variable metabolic syndrome, together with neuro-inflammation and vasculopathy [2]. Currently, the major drugs used to treat diabetes function by lowering the blood glucose levels and include metformin, acarbose, sulfonylurea, and thiazolidinediones. These drugs mainly prevent insulin resistance, increase the glucose uptake in cells, or inhibit the activities of hydrolytic enzymes such as α-glucosidase and α-amylase; however, they have severe side effects [3]. Therefore, studies on the safety and efficacy of natural compounds derived from medicinal plants have provided alternative therapeutic solutions or supplements for the treatment of diabetes diseases [4].

Lotus (*Nelumbo nucifera* Gaertn) is an aquatic angiosperm and is a traditionally used herb. It is included in the daily diet in many Asian countries [5]. Lotus plumule is commonly used in beverages because it is rich in alkaloids, sterols, and polysaccharides. Its active components have recently been widely used, owing to their various biological activities, which include their antioxidant [6], anti-inflammatory [7], hepatoprotective [8], immunoregulatory [9], antitumor [10], and antiviral [11] abilities. Some in vivo studies on lotus plumule polysaccharides (LPPS) proved its anti-diabetic potential against type 1 diabetes (T1D) via improving spontaneous inflammation, protecting pancreatic islet β cells from destruction, and alleviating subsequent chronic diabetic complications in non-obese diabetic mice [12,13]. Despite these effects, the prebiotic and insulin resistance activities of *N. nucifera* polysaccharides have not been sufficiently investigated.

Recently, several natural polysaccharides have been extracted, characterized, and used as prebiotics for promoting the proliferation of beneficial bacteria, and have shown good potential for improving the quality, flavor, and physicochemical properties of functional foods. Moreover, growing evidence has demonstrated that polysaccharides play an important role in preventing the development of type 2 diabetes (T2D) [14]. Particularly, a novel polysaccharide of *Astragalus membranaceus* has been proven to be effective in improving insulin resistance and hyperglycemia in T2D rats [14]. The polysaccharide TV LH-1, from *Trametes versicolor*, dramatically improves Akt phosphorylation and activates AMPK and glycogen synthesis in insulin-resistant HepG2 cells [15]. The insulin receptor substrate-1 (IRS1) is a well-known regulator of insulin resistance both in vitro and in vivo. The activated IRS1 acts on phosphatidylinositol 3-kinase (PI3K) to generate second messenger phosphatidylinositol (3,4,5)-trisphosphate (PIP3)-dependent kinases. This, in turn, results in interactions with, and subsequent phosphorylation of, protein kinase B (Akt) to increase glucose uptake [16,17]. Besides, polysaccharides directly present anti-diabetes activities; some polysaccharide and natural nanoparticles have been developed as oral drug delivery systems for insulin delivery [18,19]. Therefore, ideally, the combination of polysaccharide and bioactive nanoparticles could be an effective and safe therapy for diabetes.

Therefore, in this study, we investigated the effect of *N. nucifera* polysaccharides (NNP) on the enhancement of insulin sensitivity in insulin-resistant HepG2 cells via the IRS1/PI3K/AkT signaling pathway, with metformin as the positive control drug. Moreover, we investigated the prebiotic effect of NNP. The correlation between the structure and biological activities of NNP could provide experimental data as a reference for further studies and aid in the development of an effective strategy for helping patients with T2D.

## 2. Materials and Methods

### 2.1. Extraction and Purification of N. nucifera Polysaccharides

Fresh lotus seeds were dried using intermittent hot air drying with 2.0 m/s at 65 °C for 72 h. The lotus plumule was separated and milled into a powder form. The dried powder was pre-treated with anhydrous ethanol at 50 °C to remove pigments and other small alcohol soluble molecules. The pre-treated powder (200 g) was mixed with distilled water (1:10, *w*/*v*) at 92 °C for 5 h. The extract (supernatant) was collected by centrifugation at 900× *g* for 15 min. The remaining powder was mixed again with distilled water at 92 °C for 5 h and the extract was collected by centrifugation. Supernatants from both extraction steps were combined, concentrated under reduced pressure at 55 °C, and precipitated with three volumes of ethanol (95%, *v*/*v*) for 24 h at 4 °C. The precipitate was separated by centrifugation (900× *g*, 15 min) and deproteinization by employing the Sevag reagent (chloroform/n-butanol 4:1, *v*/*v*) to obtain NNP. After filtration using a Durapore^®^ PVDF membrane (0.45-μm, Millipore, Bedford, MA, USA), NNP were fractionated using gel permeation chromatography with a DEAE-cellulose DE-52 column (2.6 cm × 40 cm) and eluted with 0, and 0.5 mol/L NaCl solutions at a flow rate of 1 mL/min. The fractions were marked as NNP-1 and NNP-2 and assayed using the phenol–sulfuric acid method. The main fraction NNP-2 was collected, lyophilized, and used for further characterization. 

### 2.2. Characterization of NNP-2

#### 2.2.1. Determination of the Purity and Composition of NNP-2

The total sugar content of the polysaccharide fractions was determined by the phenol–sulfuric acid method using D-glucose as the standard [20]. Uronic acid content was measured according to the method used by Blumenkrantz and Asboe-Hansen [21] using D-glucuronic acid as the standard. Protein content was determined by the Bradford method [22] using bovine serum albumin as the standard. The content of sulfate was measured according to the method established by Dodgson and Price [23] using Na_2_SO_4_ as the standard.

#### 2.2.2. Determination of the Average Molecular Weight of NNP-2

The average molecular weights (Mw) of NNP-2 were determined using a high-performance liquid chromatograph (HPLC) (LC-10 AD Shimadzu, Tokyo, Japan) equipped with an Asahipak NH2P-50 4E column (250 × 4.6 mm, Showa Denko KK, Tokyo, Japan) and a refractive index detector (RID-10A, Shimadzu). The NNP-2 were dissolved with distilled water, filtrated, then injected into the column. The column temperature was 25 °C and the polysaccharide solution was eluted with 0.1 M NaNO_3_ solution as the mobile phase, at a flow rate of 0.5 mL/min. The molecular weight of NNP-2 was estimated from standard calibration curve of dextran (Mw: 5, 25, 150, 225, 600 kDa).

#### 2.2.3. Monosaccharide Composition 

Monosaccharide composition of the sample was analyzed by a gas chromatography-mass spectrometer (6890/5973N-GC/MSD, Agilent Technologies, Santa Clara, CA, USA), according to our previous paper [24]. Briefly, 10 mg of NNP-2 was dissolved in 10 mL of 3 M trifluoroacetic acid (TFA) and hydrolyzed at 100 °C for 3 h. The trimethylsilylated derivatives were cooled in an ice bath. Then, 1 mL of aqueous methanol was added into the reaction, followed by evaporation at 55 °C; this was repeated three times. The remaining fraction was dissolved in 1 mL of ionic water. The monosaccharides released were derived using a 0.3 M PMP methanolic solution. The aqueous layer was filtered before analyzing using HPLC–UV on a HP-5 ms capillary column and an ultraviolet detector at a flow rate of 1.0 mL/min at 30 °C. The mobile phases comprised 0.1 M phosphate buffer (pH 6.8) and acetonitrile (83:17, *v*/*v*). The reference monosaccharides with different concentrations were also derived using PMP. The molar ratio of each monosaccharide from NNP-2 was estimated according to the peak area of the reference.

#### 2.2.4. Fourier Transform Infrared (FT-IR) Analysis

The FT-IR spectra of NNP-2 were determined using a VERTEX 70 v Fourier transform infrared spectrometer (Bruker, Ettlingen, Germany) for functional group analysis. Dried NNP-2 (1 mg) was ground with 100 mg of spectra grade potassium bromide powder and pressed into tablets, and detected with FT-IR spectra in the mid-infrared range of 4000–400 cm^−1^.

### 2.3. In Vitro Prebiotic Activity of NNP

Prebiotic activity was assessed using the in vitro method with two strains of commercial probiotics viz. *Bifidobacterium adolescentis* (ATCC 15703) and *Lactobacillus acidophilus* (NCFM^®^). Analyses of prebiotic potential were performed according to previously described methods [25] with some modifications. Long-chain inulin (average DP ≥ 23, Orafti^®^ HP Inulin Powder, Quadra Chemicals, Burlington, ON, Canada), was used as the positive control, and MRS broth w/o glucose (LiofilChem Diagnostic Ltd., Roseto d’Abruzzi, Italy) was used as the control. The active probiotic strains were transferred to MRS broth and incubated at 37 °C for 24 h. Bacterial cells were collected by centrifugation (5000× *g*, 10 min) and adjusted with glucose-free MRS medium to obtain an adequate number of cells for each inoculum (1 × 10^6^ CFU/mL). Then, glucose-free MRS medium containing 0.5, 1, and 2% *w/v* carbon source (NNP-2 or inulin) was added and incubated anaerobically at 37 °C for 2 days. The bacterial counts and pH of medium were evaluated at different time intervals.

### 2.4. α-Glucosidase Inhibitory Assay

The α-glucosidase inhibitory assay was performed according to the method of Zhang et al. [26], with slight modifications. Briefly, 100 μL of α-glucosidase (0.5 units/mL) was mixed with 100 μL of different concentrations of NNP-2 (0.1–3 mg/mL) in 96-well plates and incubated at 37 °C for 10 min. Subsequently, 100 μL of 5 mM 4-Nitrophenyl β-D-glucopyranoside (PNPG) was added and the reaction mixture was incubated at 37 °C for 30 min. After incubation, the reaction was stopped by heating at 100 °C for 10 min, and absorbance was recorded at 405 nm. Metformin was used as a positive control.

### 2.5. HepG2 Cell Culture and Cell Viability Assays

The human hepatoma cell line, HepG2 (KCLB No. 58065) was obtained from the Korea Cell Line Bank (Seoul, Korea). Cells were cultured in Dulbecco’s modified Eagle’s medium (DMEM, Gibco, Grand Island, NY, USA) supplemented with 10% fetal bovine serum (FBS, Thermofisher, Scientific, Waltham, MA, USA), 10 μg/mL streptomycin, and 100 U/mL penicillin, at 37 °C in a 5% CO_2_ atmosphere. In all experiments, the cells were cultured to reach 80–90% confluence, and then cells were seeded into 96-well plates at a density of 1 × 10^4^ cells/well. After 24 h, the culture medium was removed and cells were incubated with the indicated dose of NNP-2 under the same conditions for 24 h. Then, 1 mg/mL methylthiazole tetrazolium in DMEM was added to each well to achieve a total reaction volume of 220 μL for further incubation for 4 h. The supernatant was removed, and 150 μL of dimethyl sulfoxide was added to each well. After shaking for 15 min, the amount of purple formazan formed was assessed by measuring the absorbance at 490 nm. Cell viability was determined as a percentage of viable cells in the NNP-2 treated group versus that in the untreated group.

### 2.6. Anti-Insulin Resistance Activity 

To prepare the insulin-resistant HepG2 cells model (model group), cells were washed with PBS and the culture medium was replaced with serum-free DMEM and 50 mM D-glucose medium supplemented with FBS (2%) and insulin (0.5 × 10^−4^ mM) for an additional 24 h. Subsequently, cells were washed with serum-free DMEM containing 50 mM D-glucose medium and treated with different concentrations of NNP-2 or metformin (10^−3^ mol/L). The cells maintained in DMEM containing 50 mM D-glucose and metformin (3.5 mg/L) were used as the control and metformin groups, respectively. After incubation for 24 h, culture supernatants were collected, and then the glucose concentration in the supernatants was measured using a glucose colorimetric detection kit (Invitrogen, Waltham, MA, USA).

### 2.7. Reverse Transcription-Quantitative PCR (RT-qPCR) 

Total RNA was extracted from cells using the Hybrid-R^TM^ kit (GeneAll, Seoul, Korea), according to the manufacturer’s protocol, and quantified using the NanoDrop^TM^ 2000/2000c spectrophotometer (Thermo Scientific, Waltham, MA, USA). Subsequently, RNA (2 μg) in a 20-μL reaction volume was reverse-transcribed using the Prime ScriptTMTRT kit (Takara BioInc., Otsu, Japan) using random hexamers as primers. RT-qPCR was performed using the SYBR premix Ex Taq technology (Takara BioInc., Otsu, Japan) for target mRNA (Table 1) detection in a Mic qPCR cycler (Bio Molecular Systems, Queensland, Australia). The fold change in mRNA expression levels was quantified by comparative quantitation analysis (2^–ΔΔCq^ method), which was normalized to that of the housekeeping gene β-actin [27].

### 2.8. Statistical Analysis 

Results were analyzed using the IBM SPSS 19.0 statistical software (Chicago, IL, USA) and are expressed as the mean ± standard deviation (SD) of three independent replicates. Data were analyzed using one-way analysis of variance (ANOVA) followed by Duncan’s multiple range test. Results were considered statistically significant at *p* < 0.05.

## 3. Results and Discussion

### 3.1. Determination of Purity and Characterization of NNP 

The yield of the NNP-2 fraction was 83.43% relative to that of the crude polysaccharide (Figure 1a). Purity of NNP-2 was further analyzed using an UV spectrophotometer. There was no UV absorption at 260 and 280 nm (Figure 1b), indicating that NNP-2 did not contain protein and nucleic acid. According to the standard curve for dextran, the molecular weight (Mw) of NNP-2 was estimated to be 110.47 kDa (Figure 1c). As shown in Table 2, the monosaccharides found in NNP-2 were xylose, glucose, fructose, and fucose (molar ratio of 3.25:1.03:1:0.2). The monosaccharide composition of NNP-2 was similar to that of a *Nelumbo nucifera* Gaertn (Tainan, Taiwan) polysaccharide LPPS, which mainly consisted of xylose, glucose, fructose, galactose, and fucose, with a molar ratio of 33.4:25.7:22.0:10.5:8.1 [28]. The differences in the ratio of monosaccharides indicate that the structure of *N. nucifera*-derived polysaccharides may depend on the geographical origin and the preparation method.

### 3.2. Analysis of Infrared Spectroscopy Spectrum 

The molecular structure and functional group of NNP-2 was shown in Figure 2, where the strong bond near 3372.95 cm^−1^ represented O–H stretching in the hydrogen bond, indicating strong intramolecular and intermolecular interactions between the polysaccharide chains [29]. Furthermore, the weak absorption appearing at 2933.74 cm^−1^ was ascribed to C–H stretching and bending vibrations, while the band at 1655.64 cm^−1^ indicated C=O asymmetrical stretching and the one at 1407.33 cm^−1^ indicated C=O symmetric stretching [30]. The peak at 1247.39 cm^−1^ corresponded to the typical C–O–C glycosidic band vibration [31] and 1048.90 cm^−1^ represented the C–OH stretching vibration for the pyranose form of the glucosyl residue [32].

### 3.3. Prebiotic Properties of NNP-2

The effects of NNP-2 on the proliferation of *B. adolescentis* and *L. acidophilus* are depicted in Table 3. We found that the culture medium with NNP-2 as a carbon source could promote the proliferation of both *L. acidophilus* and *B. adolescentis* significantly (compared to the basal medium). There was a dose-dependent effect of NNP-2 concentration on the number of colonies. The maximum number of colonies all appeared when the NNP-2 concentration reached 2%. Compared to those on basal media, the number of colonies of *B. adolescentis* grown in media containing NNP-2 (0.5–2% *w*/*v*) was significantly increased. However, the proliferation of *B. adolescentis* upon treatment with 2% NNP-2 was reduced compared to that upon treatment with inulin (*p* < 0.05). 

### 3.4. Inhibition of α-Glucosidase by NNP-2

As shown in Figure 3a, the α-glucosidase inhibitory activity of NNP-2 increased with increasing polysaccharide concentration from 40 to 200 µg/mL and the maximum inhibition measured was 91.32% at 200 µg/mL. However, the half-maximal inhibitory efficiency (IC50) of NNP-2 was slightly lower than that of metformin, a well-known anti-diabetic drug used as a positive control. The IC_50_ of NNP-2 and metformin were estimated to be 97.32 µg/mL and 47.23 µg/mL, respectively. 

### 3.5. Effects of NNP-2 on Glucose Uptake in Insulin-Resistant HepG2 Cells

To understand the mechanism of action of NNP-2 polysaccharides in attenuating hyperglycemia, we first detected the glucose uptake in insulin-resistant HepG2 cells. We found that glucose consumption in the model groups was much lower than that of the control HepG2 cells (*p* < 0.001), indicating the successful establishment of the insulin-resistant HepG2 cell model. NNP-2 significantly increased glucose consumption in insulin-resistant HepG2 cells in a concentration-dependent manner (*p* < 0.001). Compared with in model cells, the glucose consumption in insulin-resistant HepG2 cells treated with 200 μg/mL of NNP-2 increased by 158.8%, which was lower than that of the positive control metformin (198.3%). We then detected NNP-2′s effect on the viability of HepG2 cells using MTT assays and found that the concentration range of 50–400 μg/mL had no influence (Figure 3d). Therefore, NNP-2 can promote glucose consumption and improve the insulin-resistant HepG2 cells. 

As shown in Figure 3d–f, compared with the control group, mRNA expression of IRS-1, PI3K, and AKT was significantly suppressed in insulin-resistant HepG2 cell lines (*p* < 0.01). The NNP-2 significantly, and dose-dependently, increased the relative expression of IRS1, PI3K, and Akt. These results demonstrate that NNP-2 could effectively promote the downstream IRS1/PI3K/Akt signaling pathway, resulting in enhancing insulin sensitivity in insulin-resistant HepG2 cells. 

## 4. Discussion

Lotus plumule tea is traditionally prepared in hot water, which also causes minimal damage to the polysaccharide structure. Therefore, NNP was extracted using the hot water extraction method. The Mw of NNP-2 is smaller than that reported in a previous study [12]. The monosaccharide composition of NNP-2 was similar to that of a *Nelumbo nucifera* Gaertn (Tainan, Taiwan) polysaccharide LPPS, which mainly consisted of xylose, glucose, fructose, galactose, and fucose, with a molar ratio of 33.4:25.7:22.0:10.5:8.1 [28]. The differences in ratio of monosaccharides indicate that the structure of *N. nucifera*-derived polysaccharides may depend on their geographical origin and preparation method. The FTIR analysis showed the typical absorption of polysaccharides. The results also confirmed all previous data based on purity and monosaccharide composition analysis.

Furthermore, the number of *L. acidophilus* colonies on the medium containing NNP-2 (>0.5%) was significantly greater than that on the glucose-free medium and on the medium containing insulin, while there was no difference in the number of colonies when compared to the medium with glucose. These results indicated that NNP-2 could be a good substrate for *Lactobacillus*, and the prebiotic activity of NNP-2 on the proliferation of different probiotics in vitro varied. Monosaccharide composition was the main factor leading to prebiotic-like effects of polysaccharide [33,34]. In a recent paper, Gibson [35] asserted that higher fructose, glucose, galactose, and xylose contents in particular polysaccharides are associated with the modulation of the gut microbiota composition by increasing the diversity of microorganism species and SCFE concentrations. *L. acidophilus* and *B. adolescentis* have been commercially introduced in probiotic products that have different clinical effects [36]. Compared to the basal medium, the polysaccharide NNP-2 from lotus plumule showed higher response in the growth of these probiotics. The results were in accordance with Lin et al. [37] and Iliev et al. [38], who reported xylose-rich polysaccharides were able to be fermented by *Lactobacillus* spp. and *Bifidobacterium* spp. to convert to energy source. We speculated that the addition of NNP-2 to foodstuffs could be used as a “stimulator” to promote the growth of the *Lactobacillus* strains in gut microbiota. However, further studies are needed to elucidate their mechanism of action on prebiotic-like effects of *N. nucifera* polysaccharides. 

α-glucosidase is a key digestive enzyme that catalyzes hydrolysis of starch to α-glucose [39]. α-glucosidase inhibitors are an effective strategy in reducing post-prandial hyperglycemia. For the inhibition of mammalian α-glucosidase, metformin is the first line therapy, and is more potent against α-glucosidase than α-amylase [40]. Many researchers have focused on seeking effective α-glucosidase inhibitors from natural products with negligible adverse effects to develop functional foods against diabetes [41,42]. The α-glucosidase inhibitory activities of polysaccharides were closely related to their monosaccharide compositions, molecular weights, and type of glycosidic linkages [39]. Interestingly, the IC_50_ of NNP-2 (97.32 µg/mL) was lower than that of some reported natural polysaccharides consisting of high percentages of galactose and arabinose. According to Lv et al. [43], the polysaccharides with high content of xylose residues exhibit a significant inhibitory effect on α-glucosidase, IC_50_ range from 0.14 to 1.17 mg/mL. Wang et al. [44] reported that polysaccharides from wax apple rich in xylose and glucose (6.37 and 5.47, respectively) inhibited α-glucosidase at a concentration of 1.0 mg/mL. Moreover, the high proportion of low molecular weight polysaccharides contributed to a greater inhibition of α-glucosidase than that achieved by high molecular weight polysaccharides (107 and 197 kDa, respectively).

Numerous polysaccharides and glycoproteins have been demonstrated to improve glucose consumption via several molecular pathways involved in glucose metabolism [45]. Some polysaccharides have been investigated in clinical trials, including resistant starch, *Astragalus* polysaccharide, or ginseng polysaccharide [46,47,48]. Our preliminary study showed that aquatic plant polysaccharides can promote insulin resistance. Several pathways for glucose metabolism in the cells have been studied which include the mitogen-activated protein kinase signaling pathway, insulin signaling pathway, RAS signaling pathway, forkhead box O signaling pathway, IRS1/PI3K/Akt signaling pathway, and insulin resistance [49]. We wanted to further confirm whether the increase in glucose consumption upon NNP-2 treatment in insulin-resistant HepG2 cells was due to the inhibition of inflammation. Following treatment with metformin or NNP-2 at different concentrations for 24 h, mRNA expression levels of insulin receptor substrate 1 (IRS1), hepatic phosphatidylinositol-3-kinase (PI3K), and protein kinase B (Akt) were examined using RT-qPCR. Metformin, a first-line therapy for type 2 diabetes mellitus, is an effective and reversible selective inhibitor of IRS1/PI3K/Akt, which affects metabolism in vivo [50]. The IRS1/PI3K/AKT signaling pathway plays a vital role in the insulin signal transduction pathway and the regulation of glucose metabolism [49], thereby regulating intracellular glycogen synthesis during conditions of excessive glucose in hepatic cells. Insulin receptor substrate 1 (IRS1), the key kinase for glucose uptake and the target of insulin resistance, is a signaling factor downstream of the insulin receptor [51]. It could receive biomolecular signals via its SH2 domains and activate phosphoinositide 3-kinase (PI3K), lead to the activation of the downstream factor protein kinase B (Akt), and ultimately affect glucose and lipid metabolism. Our results showed that the supplement of the high-dose NNP-2 polysaccharides (200 µg/mL) with inulin stimulated an increase in the gene expression of the IRS1/PI3K/Akt pathway in HepG2 cells compared to the observations in the inulin group. A previous study demonstrated that some polysaccharides, such as *Sargassum fusiforme* polysaccharide [52] and *Morchella esculenta* FMP-1 polysaccharide [53], can affect the IRS1/PI3K/AKT signaling pathway, while sulfated polysaccharides derived from brown seaweed *Undaria pinnatifida* have no effect on this pathway [54].

These findings indicate that polysaccharides from *N. nucifera* might prove valuable in biomedical and pharmaceutical industries for the manufacture of natural therapeutics for lowering blood glucose and ameliorating insulin resistance in patients with T2DM. However, further studies are necessary to identify NNP-2′s chemical structure to fully understand and realize its potential in the future. To further investigate the effect of NNP-2 on glucose metabolism, animal models of diabetes mellitus are needed.

## 5. Conclusions

Polysaccharides are good biological supplements owing to their structural characteristics. In this study, NNP-2 polysaccharides from *N. nucifera* with a typical structure consisting of major monosaccharides, such as xylose, glucose, and fructose, and proper molecular weight had marked prebiotic properties, as shown by the proliferation of *Lactobacillus*. Furthermore, NNP-2 can improve insulin sensitivity by blocking the PI3K/Akt signaling pathway.

## Figures and Tables

**Figure 1 polymers-13-01780-f001:**
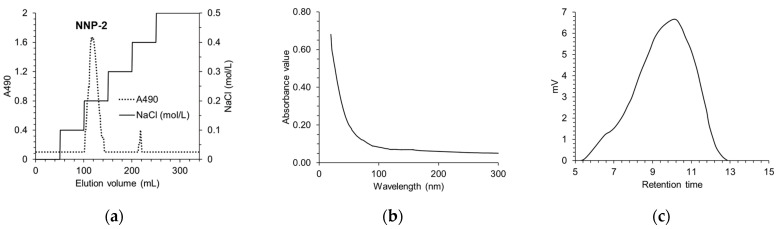
The elution curve (**a**), UV spectrum (**b**), GPC chromatogram (**c**) of NNP-2.

**Figure 2 polymers-13-01780-f002:**
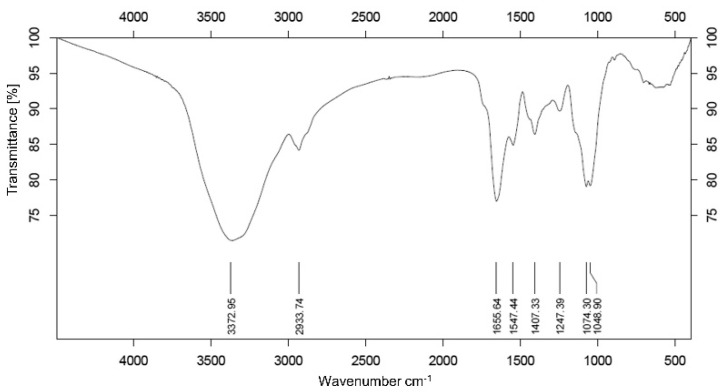
FT-IR spectrum of NNP-2.

**Figure 3 polymers-13-01780-f003:**
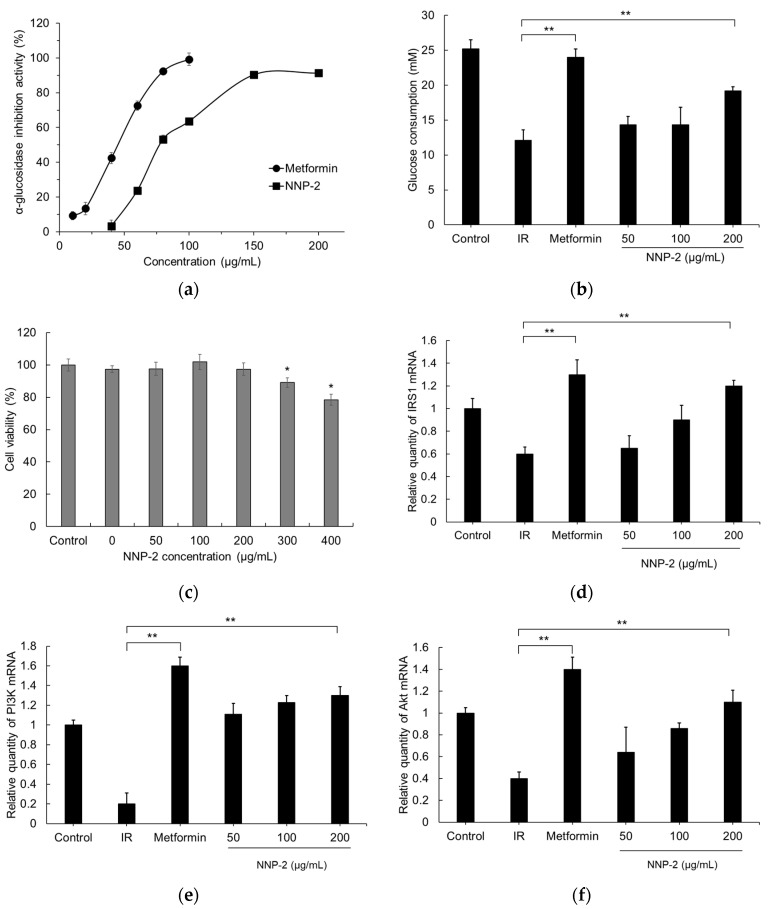
Effect of NNP-2 on inhibition of α-glucosidase activity (**a**) and insulin-stimulated glucose uptake in insulin-resistant HepG2 cells (**b**–**f**). (**b**) NNP-2 promotes glucose consumption of HepG2. (**c**) Cytotoxic effect of NNP-2. (**d**) NNP-2 regulates expression level of IRS1 mRNA. (**e**) NNP-2 regulates expression level of PI3K mRNA. (**f**) NNP-2 regulates expression level of Akt mRNA. The mRNA expressions were determined using RT-qPCR and normalized with the β-actin expression. Data are expressed as mean ± SD (*n* = 6). * *p* < 0.05 compared with the control group. ** *p* < 0.01 compared with the IR group. IR, insulin resistance cells.

**Table 1 polymers-13-01780-t001:** Primer for RT-qPCR.

Gene	Forward (5′ → 3′)	Reverse (3′ → 5′)
IRS1	GCATCGAGAAGAACAATGTG	TCCACACTCCTCGTTGTCAG
PI3K	GGTGAGGATTCAGTTGGACA	CGAGGATCCAAACCAGCTTC
Akt	AGAAGCAGGAGGACCATTTG	GCTACTCGTTCATGGTCACG
β-actin	TGCGAGAGGCATCCTCACCT	TACATGGCTGAGGTTTGAAG

**Table 2 polymers-13-01780-t002:** Sugar contents of NNP-2.

	Monosaccharide Compositions (mol%)							
Sample	Galactose	Mannose	Fucose	Xylose	Glucuronic Acid	Fructose	Glucose	Arabinose
NNP-2	-	0.03	0.2	3.25	-	1	1	-

**Table 3 polymers-13-01780-t003:** Bacterial counts of commercial probiotics at 37 °C for 24 h on MRS basal medium culture.

	Colony Forming Units (log CFU/mL) ^1^							
	Basal Medium	Glucose	NNP-2 Concentration (%)			Inulin Concentration (%)		
*Lactobacillus acidophilus*	0	0.76 ± 0.08 ^a^	0.28 ± 0.06 ^d^	0.68 ± 0.17 ^ab^	0.73 ± 0.09 ^a^	0.28 ± 0.05 ^d^	0.53 ± 0.08 ^c^	0.64 ± 0.02 ^b^
*Bifidobacterium adolescentis*	0	0.84 ± 0.03 ^a^	0.24 ± 0.01 ^e^	0.44 ± 0.09 ^d^	0.54 ± 0.03 ^c^	0.21 ± 0.06 ^e^	0.55 ± 0.03 ^c^	0.69 ± 0.03 ^b^

^1^ Data are expressed as mean ± SD (*n* = 3). Significant differences (*p* < 0.05) are indicated with different letters.

## Data Availability

The data presented in this study are available on request from the corresponding author.

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
