# Peer review of "Polysaccharide Derived from Nelumbo nucifera Lotus Plumule Shows Potential Prebiotic Activity and Ameliorates Insulin Resistance in HepG2 Cells"

_polymers, 2021, doi:10.3390/polym13111780_

Round 1

Reviewer 1 Report

Paper titled (Polysaccharide derived from Nelumbo nucifera lotus plumule shows potential prebiotic activity and ameliorates insulin resistance in HepG2 cells) by Le et al. demonstrated the separation of polysacchardies from NN species with insulin sensitizing activity. They claimed amelorative effect for the polysacchardies on PI3k/Akt signaling.

The main concern about this study is unclarity about the role of PI3k/Akt pathway.

What is the role in insulin resistance?

No previous references are shown in INTRO to rationalize selecting this pathway.

Authors also did not take a cut off decision on what is the role of the polysaccharide ? should they (inhibit or promote the pathway ) and hence improve insulin sensitivity.

Authors usually avoid this and write (ameliorate) and not say claerly whether it is inhibition or stimulation.

Please  go theough these references and consult a biologist or an expert to explain this pathway & intorduce it & discuss it

Also find the following references: I think they will be useful

https://link.springer.com/article/10.1007/s11010-011-0799-0

https://www.ahajournals.org/doi/full/10.1161/01.ATV.0000209500.15801.4e

https://pubs.acs.org/doi/abs/10.1021/acs.jafc.5b01238

Author Response

Dear Reviewer, 

Thank you for your review. We attached our responses to your comments. Please check the attached file.

Reviewer 2 Report

I read an interesting research article entitled Polysaccharide derived from Nelumbo nucifera lotus plumule shows potential prebiotic activity and ameliorates insulin resistance in HepG2 cells to Polymers Journal.  

The concept of the manuscript is novel, fits and suitable to publish in to Polymers Journal. This manuscript is generally well written and clearly presented however still needs to address many comments, and thus require substantial major revision before its acceptance.

  • In abstract authors should mention the importance of research work in one or two sentences.
  • Provide a nice graphical abstract representing the research work.
  • In the introduction section, write the novelty of the work and the problem statement clearly. Authors should discuss the nanotechnology approaches for the possible application in the treatment of diabetes for this Refer and cite Nanomaterials 10 (8), 1457, 2020; Artificial cells, nanomedicine, and biotechnology 46 (1), 211-222, 2018. More discussion of polysaccharides is essential
  • The detailed discussion about the novelty, significance of your research work and research gap relative to the literature is essential. Give detailed research objectives at the end of introduction not the repetition of abstract.
  •        Substantial discussion on quantitative data of diabetes and their treatment methods is required.
  • Give values to the peak of FTIR. Substantial discussion of peaks and their comparison with the literature is expected during revision.
  • For figure and table captions give all details which is quite expected. Don't use any abbreviation.
  • Why was alpha amylase activity not studied?
  • Have authors checked the stability of obtained polysaccharides.? Give details
  • This manuscript lacked substantial discussion of results woithe the literature authors should concentrate on this during revision.
  • Write the practical applications and future research perspectives and challenges by adding a new section before conclusions
  • The conclusion of the study is not discussed with the specific output obtained from the study, it could be modified with precise outcomes with a take home message.
  • English and grammar mistakes are present. The author should check the manuscript by native English Speaker to improve the quality of the manuscript.

Author Response

(The authors gave the same response as above.)

Round 2

Reviewer 1 Report

This reviewer does not think that the authors have revised the manuscript correctly

Hence, I do  not recommend publication

Reviewer 2 Report

The authors have substantially revised the manuscript according to the comments,

The present form of the manuscript can be accepted for publication.